

# Carbon isotopic signature of coal-derived methane emissions to atmosphere: from coalification to alteration

Giulia Zazzeri[1], Dave Lowry[1], Rebecca E. Fisher[1], James L. France[1,2], Mathias Lanoisellé[1], Bryce F.J. Kelly[4], Jaroslaw M. Necki[3], Charlotte P. Iverach[4], Elisa Ginty[4], Miroslaw Zimnoch[3], Alina Jasek[3] and Euan G. Nisbet[1].

[1]Royal Holloway University of London, Egham Hill, Egham, Surrey TW20 0EX

[2]University of East Anglia, Norwich Research Park, Norwich, Norfolk NR4 7TJ

[3]AGH-University of Science and Technology, Al.Mickiewicza 30 Kraków, Poland

[4]Connected Waters Initiative Research Centre, UNSW Australia

*Correspondence to*: Dr Giulia Zazzeri (Giulia.Zazzeri.2011@live.rhul.ac.uk)

Prof Euan Nisbet (e.nisbet@es.rhul.ac.uk)

**Abstract**. Currently, the atmospheric methane burden is rising rapidly, but the extent to which shifts in coal production contribute to this rise is not known. Coalbed methane emissions into the atmosphere are poorly characterised, and this study provides representative $\delta^{13}C_{CH_4}$ signatures to be used in regional and global models in order to allow better apportionment of fossil fuel emissions. Integrated methane emissions from both underground and opencast coal mines in the UK, Australia and Poland were sampled and isotopically characterised. Progression in coal rank and secondary biogenic production of methane due to incursion of water are suggested as the processes affecting the isotopic composition of coal-derived methane. An averaged value of -65 ‰ has been assigned to bituminous coal exploited in open cast mines and of -55 ‰ in deep mines, whereas values of -40 ‰ and -30 ‰ can be allocated to anthracite opencast and deep mines respectively. However, the isotopic signatures that are included in global atmospheric modelling of coal emissions should be region or nation specific, as greater detail is needed, given the wide global variation in coal type.

## 1    Introduction

Methane emissions from the energy sector have been driven in recent years by the impact of a shift from coal to natural gas, in the US, in the UK and Eastern Europe, whereas in China coal production has increased in this century. Currently, the atmospheric methane burden is rising rapidly (Nisbet et al., 2014), but the extent to which shifts in coal production contribute to this rise is not known. Coalbed methane emissions into the atmosphere are poorly characterised, as they are dispersed over large areas and continue even after the mines' closure (IPCC, 2006). Methane is emitted in coal processing (crashing and pulverisation) and during the initial removal of the overburden; it can be diluted and emitted through ventilation shafts in underground coal mines, or directly emitted to atmosphere from open-cut coal mining, where releases may occur as a result of deterioration of the coal seam.

For the UN Framework Convention on Climate Change, national emissions are estimated by a "bottom-up" approach, based upon a general equation where the coal production data are multiplied by an emission factor that takes into account the mine's gassiness, which in turn is related to the depth of the mine and the coal rank (i.e. carbon content of coal) (U.S. EPA, 2013). These modelled estimates are often calculated without an error



assessment, and therefore the level of accuracy of the emissions is not known. "Top-down" assessment of
methane emissions can be made by chemical transport models constrained by atmospheric measurements
(Bousquet et al., 2006; Locatelli et al., 2013). However, this top-down approach provides the total amount of
methane emissions into the atmosphere, which has to be distributed among the different methane sources in
order to quantify each source contribution.
For methane emissions from fossil fuels (coal and natural gas), the source partitioning is mainly "bottom-up",
based on energy use statistics and local inventories, which might be highly uncertain. Conversely, the "top-
down" study of the carbon isotopic composition of methane, which is indicative of the methane origin, provides
a valuable constraint on the budget appraisal, allowing different sources in a source mix to be distinguished and
their individual strength to be evaluated (Liptay et al., 1998; Lowry et al., 2001; Townsend-Small et al., 2012).
Measurements of methane mole fractions can be complemented in atmospheric models by typical $\delta^{13}CCH_4$
signatures of the main methane sources in order to estimate global and regional methane emissions and assess
emissions scenarios throughout the past years (Fung et al., 1991; Miller, 2004; Whiticar and Schaefer, 2007;
Bousquet et al., 2006; Monteil et al., 2011; Mikaloff Fletcher et al., 2014). However, even though isotopic
values are fairly distinctive for specific methanogenic processes, the variety of production pathways and local
environmental conditions that discriminate the methane formation process leads to a wide range of $\delta^{13}CCH_4$
values. The global isotopic range for coal is very large, from -80 to -17 ‰ (Rice, 1993), but it can be narrowed
down when a specific basin is studied. While there are several studies of isotopic composition of methane
generated from coal in Australia, U.S.A. and China (Smith and Rigby, 1981; Dai et al., 1987; Rice et al.; 1989;
Aravena et al., 2003; Flores et al., 2008; Papendik et al., 2011), there is a significant lack of information about
the isotopic characterisation of methane emissions from coal mines in Europe.
The purpose of this study was to determine links between $\delta^{13}CCH_4$ signatures and coal rank and mining setting,
and to provide representative $^{13}C$ signatures to be used in regional and global atmospheric models in order to
produce more accurate methane emission estimates for the coal exploitation sector.

## 1.1    Process of coalification and parameters affecting the $\delta^{13}C$ signature of methane emissions

The process of coalification involves both biochemical and geochemical reactions. The vegetal matter firstly
decays anaerobically under water; the simple molecules derived from initial decomposition (i.e. acetate, $CO_2$,
$H_2$, $NH_4^+$, $HS^-$, long chain fatty acids) are metabolised by fermentative archaea, which produce methane via two
methanogenic paths, acetoclastic reaction or $CO_2$ reduction (Whiticar, 1999). Different pathways lead to diverse
$\delta^{13}CCH_4$ isotopic signatures - methane from acetate is $^{13}C$ enriched relative to methane from $CO_2$ reduction,
ranging from -65 ‰ to -50 ‰ and -110 ‰ to -50 ‰ respectively (Levin et al., 1993; Waldron et al., 1998). With
increasing burial and temperatures, coal is subjected to thermal maturation, which implicates more geochemical
changes.
As coalification proceeds, the carbon content increases, accompanied by a relative depletion in volatile
compounds, such as hydrogen and oxygen, emitted in the form of water, methane, carbon dioxide and higher
hydrocarbons through decarboxylation and dehydration reactions (Stach and Murchison, 1982). At higher
degrees of coalification and temperature, the liquid hydrocarbons formed in previous stages are thermally
cracked to methane, increasing the amount of methane produced (Faiz and Hendry, 2006). Peat and brown coal
represent the first stage of the coalification process. The vertical pressure exerted by accumulating sediments





converts peat into lignite. The intensification of the pressure and heat results in the transition from lignite to
bituminous coal, and eventually to anthracite, the highest rank of coal (O'Keefe et al., 2013).
During peat and brown coal stages, primary biogenic methane is formed and it is mainly dissolved in water or
released during burial, as coal is not appropriately structured for gas retention (Kotarba and Rice, 2001). At
more mature stages, thermogenic methane is produced by thermal modification of sedimentary organic matter,
which occurs at great depths and intensive heat. Following the basin uplift, methane production can be triggered
in the shallower sediments by the meteoric water inflow into the coal (secondary biogenic gas) (Rice, 1993;
Scott et al., 1994).
The isotopic signature of the methane produced during the coalification process is controlled by the methane
origin pathway (Whiticar, 1996). Thermogenic methane is isotopically enriched in $^{13}$C (> -50 ‰) compared to
biogenic methane, as methanogens preferentially use the lightest isotopes due to the lower bond energy (Rice,
1993). Intermediate isotopic compositions of methane might reflect a mixing between microbial and
thermogenic gases or secondary processes. Indeed, many controlling factors co-drive the fractionation process,
and several contentions about their leverage still persist in literature. Deines (1980) asserts that no significant
trend is observed in the isotopic signature of methane in relation to the degree of coalification. Conversely,
Chung et al. (1979) observed that the composition of the parent material does affect the isotopic composition of
the methane accumulated. While the link between isotopic composition and coal rank is not that straightforward,
studies carried out in different worldwide coal seams confirm a stronger relationship between coal bed gas
composition and depth. Rice (1993), using data from Australian, Chinese and German coal beds, shows that
shallow coal beds tend to contain relatively isotopically lighter methane when compared to those at greater
depths. In the presence of intrusions of meteoric water, secondary biogenic methane, isotopically lighter, can be
generated and mixed with the thermogenic gas previously produced. Colombo et al. (1966) documented a
distinct depth correlation in the Ruhr Basin coal in Germany, with methane becoming more $^{13}$C-depleted
towards the surface zone, independently from coalification patterns. This tendency can be explained either by
bacterial methanogenesis, or by secondary processes such as absorption-desorption of methane. Also Scott
(2002), in a study about coal seams in the Bowen Basin, Australia, ascribes the progressive methane $^{13}$C-
enrichment with depth to the meteoritic recharge in the shallowest seams, associated with a higher bacterial
activity and a preferential stripping of $^{13}$C-CH$_4$ by water flow.
The migration of methane from the primary zone as a consequence of local pressure release can affect the
isotopic composition, since $^{12}$CH$_4$ diffuses and desorbs more readily than $^{13}$CH$_4$ (Deines, 1980), but the
fractionation effect due to migration is less than 1 ‰ (Fuex, 1980). A much larger variation in the isotopic
composition is associated with different methanogenic pathways (30 ‰) and thermal maturation stage (25 ‰)
(Clayton, 1998). The measurement of Deuterium, coupled with $\delta^{13}$CCH$_4$ values, would help to distinguish the
pathways of secondary biogenic methane generation, acetoclastic reactions or CO$_2$ reduction (Faiz and Hendry,
2006), but such distinction is beyond the scope of this study.
Overall, the $\delta^{13}$C values of methane from coal show an extremely wide range and understanding the processes
driving the methane isotopic composition needs to focus on the particular set of geological conditions in each
sedimentary basin.



Here we analyse the isotopic signatures of methane plumes emitted to atmosphere, from the dominant
bituminous and anthracite mines in Europe and Australia, of both deep and open cut type, to test the theories of
isotopic change due to coal rank and interaction with meteoric water.
**2     Material and Methods**
**2.1     Coal basins investigated and type of coal exploited**
**2.1.1     English and Welsh coal mines**
The major coalfields of England and Wales belong to the same stage in the regional stratigraphy of northwest
Europe, the Westphalian Stage, between roughly 313 and 304 Ma (Upper Carboniferous). Mining has ceased in
many areas. Of those remaining some are located in South Yorkshire (Hatfield and Maltby collieries) (Fig. 1),
where 50% of the coalfield's output came from the Barnsley seam, which includes soft coal overlaying a semi-
anthracitic coal and bituminous coal in the bottom portion. Seams >2m in thickness are common in the southern
half of the Yorkshire coalfield (IMC Group Consulted Limited, 2002). Upper Carboniferous coal measures
typical of Yorkshire extend into the East Midlands coalfields. The exposed coalfields are found in the west of
this belt, where the coal measures outcrop (roughly from Nottingham to Bradford and Leeds via Chesterfield,
Sheffield and Barnsley).
The coal exploited in Daw Mill Colliery is part of the Warwickshire Coalfield, from a seam which varies in
thickness from 6.6 to 7.5 m (IMC Group Consulted Limited, 2002).
The coal of the South Wales basin exhibits a well-defined regional progression in rank, which varies from
highly volatile bituminous coal in the south and east margin to anthracite in the north-west part, and the main
coal-bearing units reach 2.75 km in thickness toward the south-west of the coalfield (Alderton et al., 2004). The
coal is now preferentially extracted in opencast mines, as the extensive exploitation of the coal has left the
accessible resources within highly deformed structures (e.g. thrust overlaps, vertical faults) that cannot be
worked by underground mining methods (Frodsham and Gayer, 1999). Emissions from two deep mines, Unity
and Aberpergwm, were investigated, where mine shafts reach depths up to approximately 750 m
(http://www.wales-underground.org.uk/pit/geology.shtml).
**2.1.2     Upper Silesian Coal Basin in Poland**
The Upper Silesian Coal Basin extends from Poland to the Czech Republic and is one of the largest coal basins
in Europe, with an area of ~7400 km (Jureczka and Kotas, 1995). The Silesian Region of Poland is estimated to
be responsible for significant methane emissions, in the range of 450-1350 Gg annually (Patyńska, 2013). The
upper Carboniferous coal-bearing strata of this region are associated with gas deposits of both thermogenic and
microbial origin, and the methane content and spatial distribution are coal rank related (Kędzior, 2009) - i.e. the
sorption capacity of coal in the basin is found to increase with coal rank. Most of the methane generated during
the bituminous stage in the coalification process has escaped from the coal source following basin uplift during
Paleogene (Kotarba and Rice, 2001) and diffused through fractures and faults occurring in tectonic zones. The
late-stage gas generated by microbial reduction of $CO_2$ in the coal seams at the top of the Carboniferous
sequence accumulated under clay deposition in the Miocene (Kędzior, 2009).



### 2.1.3    Australia: Hunter Coalfield (Sydney Basin)

The Hunter Coalfield is part of the Sydney Basin, on the east coast of New South Wales in Australia, and consists of 3 major coal measures. The deepest is the early Permian Greta Coal Measures, which is overlain by the late Permian Whittingham Coal Measures and upper Newcastle Coal Measures. Throughout the Hunter coalfield all sedimentary strata are gently folded, and the same coal seam can be mined at the ground surface and at depths of several hundred meters. In the region surveyed both the opencast and underground mines are extracting coal from the Whittingham coal measures, which are generally high volatile bituminous coals, although some medium to low bituminous coals are extracted (Ward and Kelly, 2013). The mining operation is on a much larger scale than in the UK - the total coal production for the Hunter Coalfield was 123.63 Mt in 2011, of which 88.24 Mt was saleable (State of New South wales, 2013)-. Several studies have attested to the dispersion of thermogenic methane formed at higher degrees of coalification (i.e. high temperatures and pressures) during the uplift and the subsequent erosion of the basin, followed by the replenishment of the unsaturated basin with more recently formed methane of biogenic (Faiz and Hendry, 2006; Burra, 2010). Gas emplacement is related to sorption capacity of coal strata, in particular to the pore pressure regime, which is influenced by the local geological features (compressional or extensional) of the basin. The south of the Hunter Coalfield is characterized by higher gas content and enhanced permeability than the northern area, with a large potential for methane production, mainly biogenic (Pinetown, 2014).

### 2.2    Sampling and measurement methodology

For isotopic characterisation of the methane sources, integrated methane emissions were assessed through detection of the offsite downwind plume. In fact, even when emissions are focused on defined locations, such as vent pipes in underground mines, the methane provenance cannot be localised, since most of collieries are not accessible. For sample collection and measurements of methane emissions downwind of coal mines in the UK and Australia the mobile system described by Zazzeri et al. (2015) has been implemented. The system utilises a Picarro G2301 CRDS (Cavity Ring-Down Spectroscopy) within the survey vehicle, for continuous $CH_4$ and $CO_2$ mole fraction measurements, and a mobile module including air inlet, sonic anemometer and GPS receiver on the roof of the vehicle. The entire system is controlled by a laptop, which allows methane mole fractions and the methane plume outline to be displayed in real time on a Google Earth platform during the survey to direct plume sampling. When the plume was encountered, the vehicle was stopped and air samples collected in 3L Tedlar bags, using a diaphragm pump connected to the air inlet. Samples were taken at different locations along the plume transect in order to obtain a wide range of methane mole fractions and isotopic signatures in the collected air. The Upper Silesian basin was surveyed with a Picarro 2101-i measuring continuous $CO_2$ and $CH_4$ mole fractions and $\delta^{13}CCO_2$ isotopic ratio with a precision of 0.3 ‰ in 5 minutes. Samples were collected on site for analysis of $\delta^{13}CCH_4$ isotopic ratio.

The carbon isotopic ratio ($\delta^{13}C$) of bag samples was measured in the greenhouse gas laboratory at RHUL (Royal Holloway University of London) in triplicate to high precision (±0.05 ‰) by continuous flow gas chromatography isotope ratio mass spectrometry (CF GC-IRMS) (Fisher et al., 2006). $CH_4$ and $CO_2$ mole fractions of samples were measured independently in the laboratory with a Picarro G1301 CRDS analyser, calibrated against the NOAA (National Oceanic and Atmospheric Administration) WMO-2004A and WMO-X2007 reference scales respectively. The $\delta^{13}CCH_4$ signature for each emission plume was calculated using the





Keeling plot approach, according to which the $\delta^{13}C$ isotopic composition of samples and the inverse of the
relative mole fractions have a linear relationship, whose intercept represents the isotopic signature of the source
(Pataki et al., 2003). Source signatures were provided with the relative uncertainty, computed by the BCES
(Bivariate Correlated Errors and intrinsic Scatter) estimator (Akritas and Bershady, 1996), which accounts for
correlated errors between two variables and calculates the error on the slope and intercept of the best
interpolation line. Mole fraction data and co-located coordinates were used to map the mole fraction variability
using the ArcGIS software.

## 198    3    Results and Discussion

While most of the emissions from deep mines come specifically from ventilation shafts, which are point
sources, emissions from open-cut mines are wide-spread, and difficult to estimate. However, the objective of
this study is not the quantification of emissions, but the assessment of the overall signature of methane released
into the atmosphere, made through the sampling of integrated emissions from the whole area. Therefore, even
though onsite access to collieries was not possible, by driving around the contiguous area, methane emissions
could be intercepted and their mole fractions measured. Table 1 summarises $\delta^{13}CCH_4$ signatures of all the coal
mines and coal basins surveyed with the Picarro mobile system. The English and Welsh coal mines surveyed are
shown in Fig. 1.

### 207    3.1    English coal mines

Emissions from Hatfield colliery, one of the few UK deep mines still open at the time of this study, were
investigated on 10[th] July and 26[th] September 2013. The main seams that have been worked since 1920's are the
Barnsley, the Dunsil and the High Hazel seams, at approximate depths of 401-365 m 414-376 and 313-284 m
respectively (Hill, 2001). Mole fractions over 7 ppm were recorded during both surveys while transecting the
plume (Fig. 2). The methane desorbed from coal heaps within the colliery area and methane vented from shafts
might explain the relatively high mole fractions measured. Keeling plots based on the samples collected (4 on
the first and 5 on the second survey) give intercept values of -48.3 ±0.2 and -48.8 ±0.3 ‰ (2SD).
Maltby colliery is one of the largest and deepest mines in England, reaching 991 m in depth, and was closed in
March 2013. High volatile bituminous coal was extracted from the Barnsley and Parkgate seams (McEvoy et al.,
2006). The coal mine methane was extracted and used for electricity, but over 3 ppm mole fractions were
detected in ambient air during both surveys in July and September 2013, giving evidence of methane releases
from the recovery system and ventilation shafts. The Keeling plot intercept based on the samples collected in
July and in September are -45.9 ±0.3 and -45.4 ±0.2 ‰ (2SD) respectively, ~3 ‰ heavier for Hatfield Colliery
on the same sampling days. Both source signatures are in good agreement with the value of -44.1 ‰ observed
for desorbed methane of the Barnsley coal seam, in a study conducted by Hitchman et al. (1990b).
Kellingley colliery is a still open deep mine situated in North Yorkshire, which has exploited the Beeston and
Silkstone coal seams, characterised by a high quality, low sulphur coal used for coke manufacture. A surface
drainage plant for electrical power generation has been implemented (Holloway et al., 2005), and methane
releases from the power plant might justify the mole fractions peak of 9 ppm that was observed while driving on


the north side of the coal mine on 10th July 2013. The Keeling plot analysis of the samples collected indicates a
source signature of -46.5 ±0.3 ‰ (2SD).
Thoresby mine is located in Nottinghamshire and is currently exploiting the Parkgate seam at about 650 m
underground. The highest mole fractions detected approached 5 ppm, and 9 samples were collected, giving a
source signature of -51.2 ±0.3 ‰ (2SD).
The Daw Mill colliery, before its closure in March 2013, was Britain's biggest coal mine, working the
Warwickshire Thick seam, which lies between 500 and 1000 m in depth. The estimated methane content of this
seam is low (typically about 1.7 $m^3$/tonne) and, furthermore, a ventilation system is implemented, so that the
coal mine methane potential is curtailed (Drake, 1983). During the sampling, the highest methane mole fractions
(≈5 ppm) were recorded close to the edge of the colliery, whereas, driving downwind of the site at further
distances, only background values were measured. Keeling plot analysis reveals a source signature of -51.4 ±0.2
‰ (2SD).
**3.2    Welsh coal mines**
Methane emissions from coal mines in Wales were sampled in order to characterise methane releases from a
different rank of coal. The area investigated extended from Cwmllynfell to Merthyr Tydfil (Fig. 1). The South
Wales coalfield is estimated to have the highest measured seam gas content in the UK (Creedy, 1991).
**3.2.1    Deep Mines**
The deep mine Aberpergwm, which closed in December 2012, did not operate coal mine methane schemes and
methane was vented up to the surface as part of standard operation systems (Holloway et al., 2005). That is
consistent with methane mole fractions peaks of 6 ppm observed when approaching the colliery. Air samples
were also collected near Unity deep mine, Wales' largest drift mine, which reopened in 2007 and is located in
the town of Cwmgwrach, only one mile away from Aberpergwm. Isotopic source signatures of 33.3 ±1.8 ‰ and
-30.9 ±1.4 ‰ result from the Keeling plots based on samples collected respectively near Aberpergwm and Unity
colliery, both highly $^{13}$C enriched relative to all English collieries.
**3.2.2    Opencast Mines**
The Picarro mobile system was driven around opencast mines at Cwmllynfell and Abercrave, in the Swansea
Valley. Up to 3 ppm methane mole fractions were recorded near the two mines, which were closed in the 1960s
as drift mines and are currently exploited as opencast mines. Our measurements confirm that they are still
emitting methane, albeit at levels which are not significantly above background. $^{13}$C signatures between -41.4
±0.5 and -41.2 ±0.9 ‰ (2SD) result from isotopic analysis of samples collected downwind of the opencast
mines, approximately 10 ‰ lower than the isotopic signature characterizing Welsh deep anthracite mines.
**3.3    Polish coal mine**
A Picarro mobile survey of the Upper Silesian basin took place on 10th June 2013 and 12 air samples were
collected for isotopic analysis. All the mines in this area are deep mines, exploiting the coal at depths ranging
from 300 to 900 m. The Keeling plot analysis includes 8 samples collected around the area of Radoszowy,
downwind of the KWK Wujek deep mine shafts (white stars in Fig. 4). Methane mole fractions in the range of





3-5 ppm were measured in the majority of the area of Katowice and over 20 ppm mole fractions were detected
when transecting the plume originating from the exhaust shafts, which confirms the high level of methane that
the mine contains. A source signature of -50.9 ± 0.6 ‰ (2SD) was calculated by Keeling plot analysis, which is
consistent with the values obtained for the deep mined English bituminous coal.
**3.4    Australia: Hunter Coalfield**
On 12th and 18th March 2014 12 samples in total were collected along the route in the Hunter Coalfield, where
the bituminous coal strata of the Sydney Basin are extracted both in opencast and underground mines. The
methane plume width was in the range of 70 km (Fig. 5a). A maximum mole fraction of 13.5 ppm was measured
near a vent shaft associated with the Ravensworth underground mine (see white star in Fig. 5b), which exploits
the Lemington, Pikes Gully, Lidell (Upper and Middle) and Barret Seams, the deepest of which is the Barret
seam, with a maximum overburden depth of ~350 m (GSS Environmental, 2012). The source signature
calculated by the Keeling plot analysis based on all the samples collected during both surveys (grey markers in
Fig. 5c) is -66.4 ±1.3 ‰ (2SD) and this signature likely includes a mixture of methane derived both from
underground and opencast mines. Two samples collected downwind of the Ravensworth ventilation shaft (red
pushpins in Fig. 5b) fall off the Keeling plot trend for March 2014 and are not included in the calculation,
because they are likely dominated by methane from the vent shaft and are not representative of the regional
mixed isotopic signature.
In January 2016, 10 samples were collected downwind of a ventilation fan in the Bulga mine (second white star
in Fig. 5b), aerating underground workings in the Blakefield South Seam, ranging from 130 to 510 m in depth
(Bulga Underground Operation mining, 2015). The Keeling plot for these (black circles in Fig. 5c) indicates a
$\delta^{13}C$ source signature of -60.8 ±0.3 ‰ (2SD). The samples collected next to the Ravensworth ventilation shaft in
2014 fit on this Keeling plot, suggesting that the $\delta^{13}CCH_4$ isotopic signature of emissions from underground
works in the Hunter Coalfield is consistent.
**3.5    New representative $\delta^{13}CCH_4$ isotopic signatures for coal-derived methane**
The $\delta^{13}CCH_4$ isotopic values for coal have been found to be characteristic of single basins, but general
assumptions can be made to characterise coal mines worldwide. Table 2 provides the literature $\delta^{13}CCH_4$ isotopic
values characteristic of specific coal basins. The isotopic signatures of emissions from English bituminous coal
are consistent with the range of -49 to -31 ‰ suggested by Colombo et al. (1970) for in situ coal bed methane in
the Ruhr basin in Germany, which contains the most important German bituminous coal of Upper Carboniferous
age and low volatile anthracite (Thomas, 2002). Progression in coal rank might explain the value of -50 ‰ for
emissions from English underground mined bituminous coal and -30 ‰ for anthracite deep mines, followed by a
5-10 ‰ $^{13}$C-depletion caused by the incursion of meteoric water in the basin and the subsequent production of
secondary biogenic methane, resulting in -40 ‰ for methane plumes from Welsh open-cut anthracite mines.
The link between coal rank and $\delta^{13}CCH_4$ isotopic signature is appreciable in the study of UK coal mines, but
differences in the $^{13}C/^{12}C$ isotopic ratio within the same coal sequences can be ascribed to other parameters, such
as the depth at which the coal is mined and the occurrence of water incursion. In particular, emissions from
Thoresby are more $^{13}$C depleted than those measured around Maltby, although both mines exploit the Parkgate
seam, meaning that different isotopic signatures cannot be entirely linked to the coal rank. Biogenic methane





produced in a later stage due to water intrusion might have been mixed with the original thermogenic methane
formed during the coalification process.
Differences in methane emissions and their isotopic signature between opencast and deep mines have been
assessed by surveying both surface and underground mines, in the Welsh anthracite belt, and in the Hunter
Coalfield. The shallower deposits have been more exposed to the weathering and meteoric water, most likely
associated with the production of some isotopically lighter microbial methane. Mole fractions up to 2.5 ppm
were measured around opencast mines in the Hunter Coalfield, in Australia, within a methane plume of more
than 70 km width. The highest methane mole fractions were consistently measured downwind of vent shafts in
underground mines. The difference in the source isotopic signature for methane emissions between the two
types of mining in the Hunter Coalfield (from -61 to -66 ‰) and in Wales (from -31 to -41 ‰) reflects the
isotopic shift of 5-10 ‰ that has been attributed to the occurrence of secondary biogenic methane.
The $\delta^{13}$C signatures for coalbed methane emissions from the Upper Silesian basin are highly variable, with the
most $^{13}$C depleted methane associated with diffusion processes or secondary microbial methane generation
(Kotarba and Rice, 2001), but the value of -51 ‰ measured for methane emissions from the KWK Wujek deep
mine is consistent with the value of -50 ‰ inferred for emissions from English bituminous coal extracted in
underground mines.
**4    Conclusions**
By measuring the isotopic signatures of methane plumes from a representative spread of coal types and depths,
we show that the $\delta^{13}$C isotopic value to be included in regional and global atmospheric models for the estimate
of methane emissions from the coal sector must be chosen according to coal rank and type of mining (opencast
or underground). For low resolution methane modelling studies an averaged value of -65 ‰ is suggested for
bituminous coal exploited in open cast mines and of -55 ‰ in deep mines, whereas values of -40 ‰ and -30 ‰
can be assigned to anthracite opencast and deep mines respectively.
Global methane budget models that incorporate isotopes have used a $\delta^{13}$C signature of -35 ‰ for coal or a value
-40 ‰ for total fossil fuels (e.g. Hein et al., 1997; Mikaloff Fletcher et al., 2004; Bousquet et al., 2006; Monteil
et al., 2011), but, given the relative rarity of anthracite coal reserves and the dominance of bituminous coal
(http://www.transparencymarketresrch.com/anthracite-coal-mining.html), it seems likely that a global average
emission from coal mining activities will be lighter, with the -50 ‰ recorded for deep-mined bituminous coal in
Europe being a closer estimate. However, for detailed global modelling of atmospheric methane, isotopic
signatures of coal emissions should be region or nation specific, as greater detail is needed given the wide global
variation. The assignment of an incorrect global mean, or a correct global mean but inappropriate for regional
scale modelling, might lead to incorrect emissions estimates or source apportionment. The new scheme gives the
possibility for an educated estimate of the $\delta^{13}$C signature of emission to atmosphere to be made for an individual
coal basin or nation, given information on the type of coal being mined and the method of extraction.
In conclusion, high-precision measurements of $\delta^{13}$C in plumes of methane emitted to atmosphere from a range
of coal mining activities have been used to constrain the isotopic range for specific ranks of coal and mine type,
offering more representative isotopic signatures for use in methane budget assessment at regional and global
scales.



**Acknowledgements**
Giulia Zazzeri would like to thank Royal Holloway, University of London for provision of a Crossland
scholarship and a contribution from the Department of Earth Sciences from 2011 to 2014. Analysis of samples
from Poland was funded through the European Community's Seventh Framework Programme (FP7/2007-2013)
in the InGOS project under grant agreement n. 284274. Sampling in Australia was possible due to a grant from
the Cotton Research and Development Corporation.

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







| Sampling Site | Country | Sampling Date | $\delta^{13}$C Signatures [‰] | Samples number used in Keeling plots |
|---|---|---|---|---|
| Kellingley Colliery | North Yorkshire/UK | Sept-2013 | -46.5 ±0.3 | 5 |
| Maltby Colliery | South Yorkshire/UK | Jul-2013 | -45.9 ±0.3 | 3 |
|  |  | Sept-2013 | -45.4 ±0.2 | 4 |
| Hatfield Colliery | South Yorkshire/UK | Jul-2013 | -48.3 ±0.2 | 4 |
|  |  | Sept-2013 | -48.8 ±0.3 | 5 |
| Thoresby Colliery | Nottinghamshire/UK | Nov-2013 | -51.2 ±0.3 | 9 |
| Daw Mill Colliery | Warwickshire/UK | Nov-2013 | -51.4 ±0.2 | 4 |
| Cwmllynfell Colliery | Wales/UK | Oct-2013 | -41.2 ±0.9 | 4 |
| Abercrave Colliery | Wales/UK | Oct-2013 | -41.4 ±0.5 | 5 |
| Aberpergwm Colliery | Wales/UK | Oct-2013 | -33.3 ±1.8 | 3 |
| Unity Colliery | Wales/UK | Oct-2013 | -30.9 ±1.4 | 5 |
| Hunter Coalfield | Australia | Mar-2014 | -66.4 ±1.3 | 12 |
| Bulga Colliery | Australia | Jan-2016 | -60.8 ±0.3 | 10 |
| Boggabri and Tarrawonga Collieries | Australia | Mar-2014 | -55.5 ±1.3 | 5 |
| Upper Silesian Basin | Poland | Jun-2013 | -50.9 ± 1.2 | 8 |

**Table 1 $\delta^{13}$CCH$_4$ Signatures of all the coal mines and coal basins surveyed with the Picarro mobile system. Errors in**
**the $\delta^{13}$C Signatures are calculated as 2 standard deviations.**





| Site | Coal Rank | δ¹³C ( ‰) | Author | δ¹³C ( ‰) measured in this study |
|---|---|---|---|---|
| Velenje Basin, Slovenia | Lignite | -71.8 to -43.4 | Kanduč et al. (2015) | |
| Australia | From brown coal to low volatile bituminous coal. | -73.0 to -43.5 | Smith et al. (1981) | -66.4 ±1.3 to -55.5 ±1.3 |
| Powder River Basin, U.S.A. | Sub-bituminous coal | -68.4 to -59.5 | Flores et al. (2008) | |
| Queensland Basin, Australia | From Sub-bituminous to high volatile bituminous | -57.3 to -54.2 | Papendick et al. (2011) | |
| San Juan basin, New Mexico and Colorado | High-volatile Bituminous coal | -43.6 to -40.5 | Rice et al. (1989) | |
| Elk Valley Coalfield, British Colombia | Bituminous coal | -65.4 to -51.8 | Aravena et al. (2003) | |
| UK, Barnsley seam | Bituminous coal | -44.1 | Hitchman et al. (a, b) | -48.8 ±0.3 to -45.4 ±0.2 |
| Upper Silesian Coal Basin, Poland | Sub-bituminous coal to anthracite | -79.9 to -44.5 | Kotarba and Rice (2001) | -50.9 ±0.6 |
| Eastern China | Sub-bituminous to anthracite | -66.9 to -24.9 | Dai et al. (1987) | |
| Ruhr basin, Germany | Bituminous coal, Anthracite | -37 (-60 to -14) | Deines (1980) | |
| Western Germany | High-volatile bituminous to anthracite. | -70.4 to -16.8 | Colombo et al. (1970) | |
| Qinshui, China | Anthracite | -41.4 to -34.0 | Qin et al. (2006) | |
| Wales, UK | Anthracite | | | -33.3 ±1.8 to -30.9 ±1.4 |

**Table 2 Literature isotope values obtained by methane samples from boreholes and coal seams. Errors are not included in the sources.**







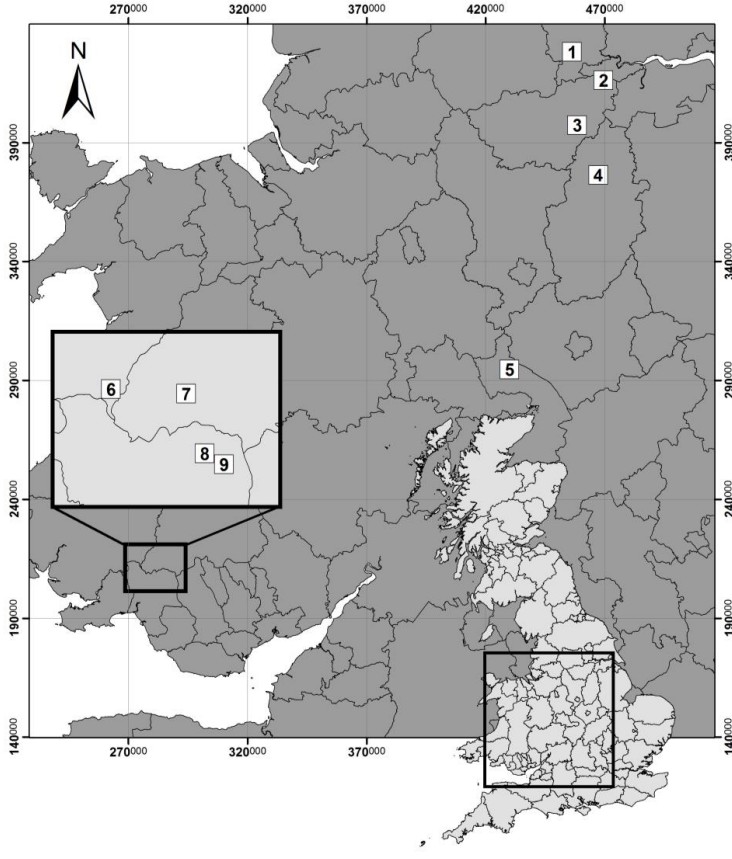


**Figure 1 Location of English coal mines (1 Kellingley, 2 Hatfield, 3 Maltby, 4 Thoresby, 5 Daw Mill) and Welsh coal mines in the anthracite belt (6 Cwmllynfell, 7 Abercrave, 8 Unity, 9 Aberpergwm). Coordinates are displayed in the National British Coordinates System.**





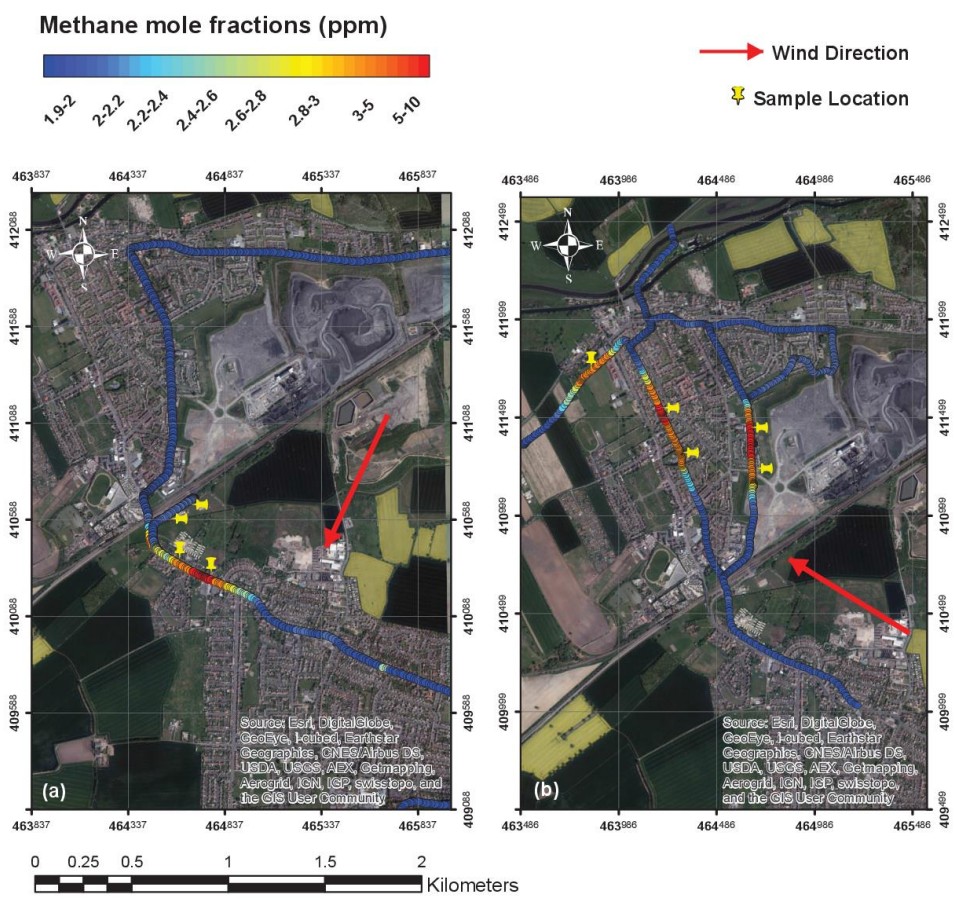


**Figure 2 ArcGIS maps of methane mole fractions recorded downwind of Hatfield Colliery on July (a) and September**
**2013 (b). Grid Coordinates are displayed in the National British Coordinate System.**



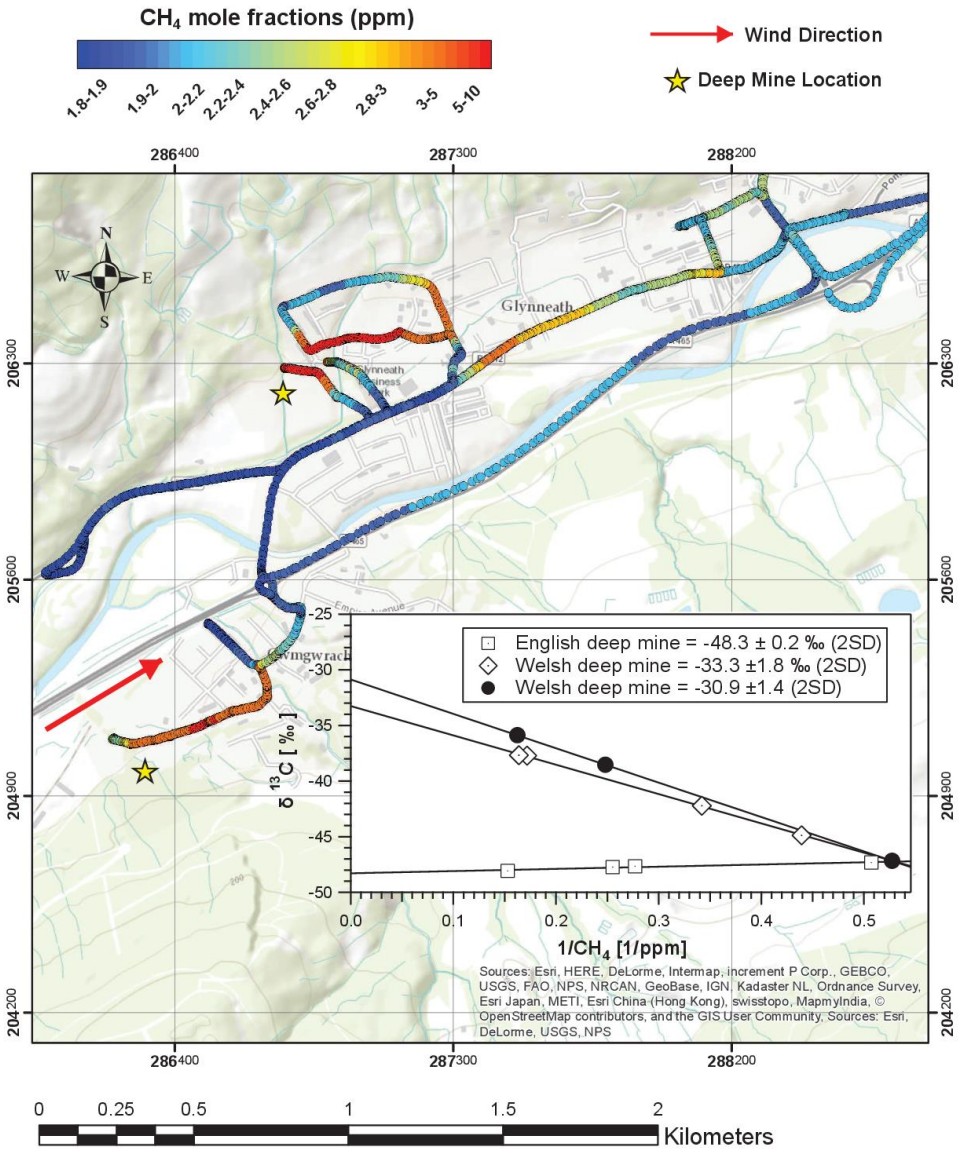


**Figure 3 ArcGIS plot of methane mole fractions recorded in October 2013 next to Aberpergwm and Unity deep mines in Wales –yellow stars. Grid Coordinates are displayed in the National British Coordinate System. The embedded graph is the Keeling plot based on samples collected downwind of one English coal mine (Hatfield Colliery) and two Welsh deep mines (Aberpergwm and Unity Colliery). Errors on the y-axis are within 0.05 ‰ and on the x-axis 0.0001 ppm$^{-1}$, and are not noticeable on the graph.**





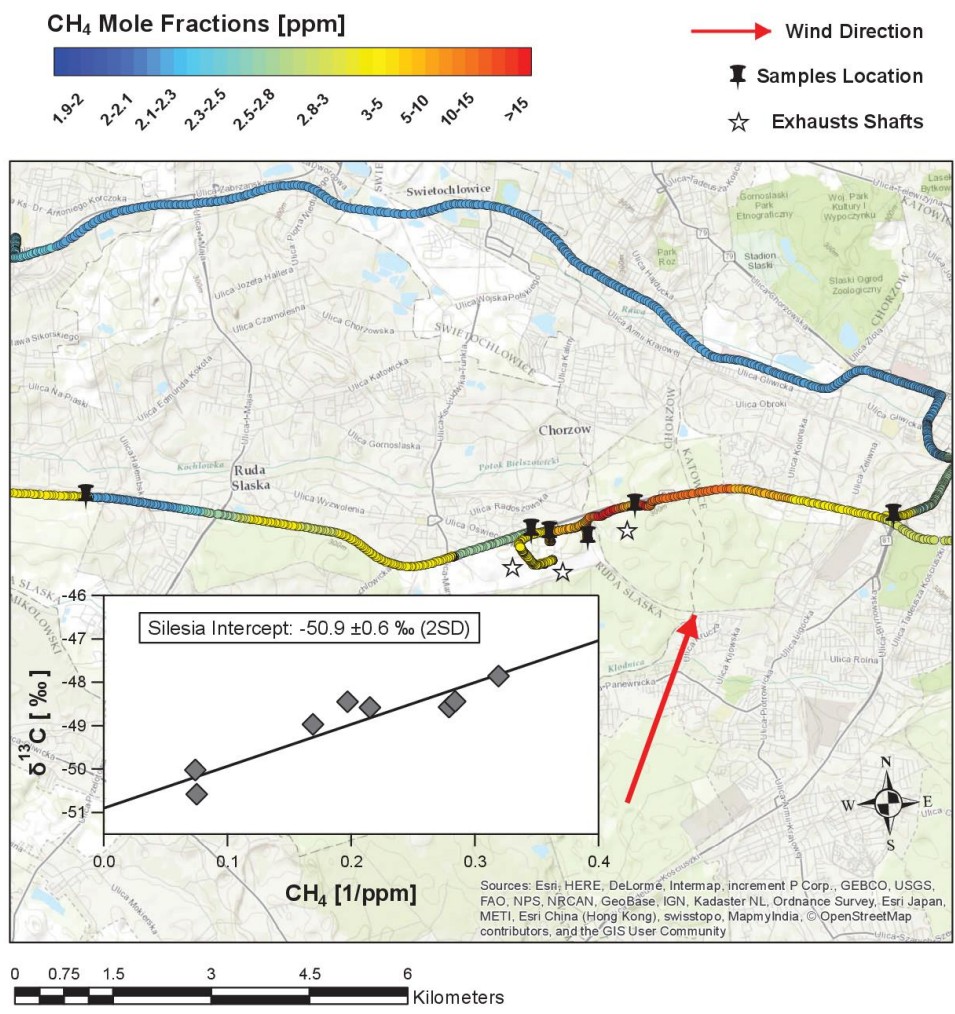

513

**Figure 4 ArcGIS map of methane mole fractions recorded in the Upper Silesian basin in June 2013. White stars represent the KWK Wujek deep mine exhausts shafts. The embedded graph is the Keeling Plot based on samples collected downwind the KWK Wujek deep mine in June 2013. Errors on the y-axis are within 0.05 ‰ and on the x-axis 0.0001 ppm⁻¹, and are not noticeable on the graph.**





518

**Figure 5 Methane mole fractions recorded in the Hunter Coalfield on 12th March 2014 (a) and during a more detailed survey of the area highlighted by the black square on 18th March 2014 (b). Keeling plot based on samples collected along the route in the Hunter Coalfield in March 2014 (grey and red markers) and near to ventilation in January 2016 (black markers) (c). Errors on the y-axis are within 0.05 ‰ and on the x-axis 0.0001 ppm$^{-1}$, and are not noticeable on the graph.**

524

525