# Peer review of "Carbon isotopic signature of coal-derived methane emissions 2 to atmosphere: from coalification to alteration"

_Atmospheric Chemistry and Physics, 2016_

## Short Comment (SC1) · 5 Apr 2016

I think this paper must stress the fact that global modeling studies have been using a d13CH4 signal from coal emission of -35‰ based on Deines 1980. However, after reading this paper, I have realized that this is a rather unrealistic value because it is only representative of anthracite mines, which are probably less abundant and more restricted to certain regions (e.g. Western Europe, Eastern China, Appalachians) than lower rank coals. I think global modelers would benefit if the information in table 2 would be accompanied by a map with the median coal mining d13CH4 signal from each country or basin. Moreover, as a kind of uncertainty analysis, the d13CH4 signal obtained could be compared to the signal calculated using alternative methods,
e.g. a Miller-Tans plot [Miller and Tans, 2003]. Although this might exceed the scope of this work, the author might want to consider providing a weighted mean coal mining d13CH4 signal (based on the source strengths per country from EDGARv4.2 or Schwietzke et al. [2014]) because most global atmospheric d13CH4 studies are done with box models.

Additionally, the information on the air sampling strategy should be extended, e.g. how many flask samples, how often were they taken, how was it chosen when to sample, are there background samples? Additionally, more information on the d13CH4 analysis should be given, e.g. reference standard (e.g. VPDB, NBS-19 material or other), calibration gas, precision of measurements).

I would also like to point out that there is a large map containing the location of UK mines but not of Australia and Poland. Adding a layer of coal seems to the map would also be helpful, if this is available.

Finally some references in the text are wrong: Mikaloff-Fletcher et al. 2014 should be 2004 Smith and Rigby 1981 should be Smith et al. 1981

---

## Referee Comment (RC1) · Anonymous Referee #1 · 3 May 2016

The manuscript of Zazzeri et al., aims at providing a representative estimate of the d13C isotopic signature of methane emissions from the coal mining sector. This is an important topic, since this signature can be used as a proxy for the contribution of the coal mining sector to the global methane emissions budget.

They determine the isotopic signature of methane emissions from coal mines in the UK, Poland and Australia, using the so-called "Keeling plot" technique on d13C-CH4 measurements in air samples taken downwind of coal mines. Based on these observations, they propose a range of source signatures that is significantly more depleted (by 15 to 20 permil) than what has been used so far in global CH4 modelling studies. This, and the fact that these findings are well supported, makes the paper very worthy

of publication.

I have no major negative comments: the objective of the study is clearly stated and the method followed is very well suited to that problem (this may actually be quite the ideal case for using the "Keeling plot" technique). The paper is well written and concise, and the supporting figures are overall appropriate. I have a few small comments, listed below, but I think that they can all be addressed within a "technical correction", after which the paper can be published.

**Minor comments:**

- One useful information that I think is missing is the importance of the methane emissions from the regions studied. Some values are provided for the Silesian mines, and some estimates of the coal production is given for the Australian ones, but how much does that represent on the world's scale, and how important are the British mines? Some literature references would suffice.

- Table 2 could be extended with information on the mine type (open, deep, active, inactive since, ..., perhaps also coordinates) and coal type.

- I don't find the map in Figure 1 very interesting: showing just the location of the mines without context information does not bring much to the paper. You could add some simplified geological map, or a map of the estimated distribution of coal mining methane emissions. Also, it is strange not to have the Australian and Silesian mines on a map!

- Are you expecting any long or short-term variability of the d13C signature within one coal mine? If you were to re-do the sampling at a different season, or in a few years, would you expect any different result?

- Many of the readers of this paper won't be geologists. Although Section 1.1 is helpful

and well written, it may help to provide some diagram showing the coal maturation process (when to expect which coal type), to be used as a reference when reading the following sections.

**Textual comments:**

- Line 26-27: what about Germany

- Line 67: . . . methanogenic path: acetoclastic . . . (semicolumn missing)

- Line 255: the measurements are not significantly above background: is that because the mine emits less, or because the emissions are spread over a larger area in such an open mine?

- L298: why would the depth affect the d13C (other than the depth/coal-rank relation?)

- L310: There is a 20-35 permil difference between the Hunter Coalfield and the Wales mines, and you are talking of a 5-10 permil shift here . . .

- L321-323: Do you have any information on the proportion of each coal/mine type in the world?

---

## Author Comment (AC1) · 12 May 2016

Dear Nuñez Ramirez,

thank you very much for your comments, they are extremely constructive.

I will certainly add more information about the $\delta$13CH4 analysis. The paper Zazzeri et al. (2015), which I refer to for the sampling strategy, explains in more detail the whole sampling methodology, but I agree that more details for a better understanding of the method can be added.

More maps will be provided, as long as they are available.

[Figure]

I think the Keeling Plot analysis is well suited in the special circumstances of our measurements. The Miller-Tans plot allows for a changing isotopic ratio of the background, it is useful when the background is not constant and there is more than one source. In our case, each sampling aimed at identifying the isotopic source signature of one source and the sample collected upwind of the source represented the background.

The calculation of a weighted mean coal $\delta 13CH4$ signal could be the scope of a different paper that could include some modelling work rather than only field measurements, which this paper is based on. However, even if all the studies of coal are located and provide details of mining and coal type, a global mean might not be representative of global coal mining as a whole, which is why we are advocating the calculation of regional averages.

Kind Regards

---

## Referee Comment (RC2) · I. Levin (Referee) · 16 Jun 2016

The authors provide a survey of stable carbon isotopic signatures of methane, emitted from coal deposits of different rank, collected around opencast mining areas and in the vicinity of vents from underground mines. They analyzed atmospheric grab samples with different elevated CH4 concentrations and used the Keeling plot method to derive mean isotopic signatures of the CH4 emissions. Their results obtained from different mining areas in England and Wales as well as in Upper Silesia, Poland, and in Australia fit well into the range of earlier studies. However, they are considerably different, i.e. more depleted, compared to the values generally used in global atmospheric methane budgeting studies. A dependency of the isotopic signature on the coal type (rank), and

also on the situation of the sampled coal deposit is found.

The study provides very valuable new data that deserve publication. However, I feel that the manuscript would largely benefit from restructuring/streamlining and focusing, as far as possible, also on a generalization of the findings. This would facilitate future use of the new data by non-specialists and make them much more valuable as input into global and regional CH4 budgeting studies.

I have a number of general points, which should be addressed by the authors in order to meet the required scientific quality for publication in ACP:

1. The results are currently presented in a very descriptive way, giving a lot of detail that makes part of the manuscript hard to read. I feel that a lot of this text, e.g. about locations, setting, would better fit into Table 1.

2. For each campaign, the authors report on the maximum CH4 signal and present detailed figures about the measured concentration distributions. However, this information is not evaluated in any way in the manuscript (e.g. by applying a dispersion model to estimate emission rates). The concentration signals largely depend on the actual meteorological situation at the time of sampling; therefore, a value of e.g. 10 ppm, which certainly confirms that there are indeed releases from the mining area, does not tell anything quantitative about the strength of the emission. I therefore feel that most of the figures are not required. Instead, in order to give the reader a feeling about the homogeneity of the Keeling-plot derived isotopic signature at each individual location and campaign, it would be more appropriate to present, perhaps in one multi-panel figure, individual Keeling plots from ALL campaigns, therewith showing the individual data points and regression lines in a standardized way (e.g. similar as Figure 5c, may be indicating also the maximum concentration in addition to 1/CH4).

3. As also indicated by referee #1, I would strongly recommend to prepare an additional figure that summarizes all results, may be including the already published values from Table 2, in a diagram showing the relation between isotopic signature and coal type.

This would be of tremendous help for people who want to use these results in their (modelling) studies. Currently, the last sentence in the Abstract leaves the reader/user somehow alone with the problem that rather no representative or at least improved input value for d13C of CH4 from coal is available. It looks to me that with the background knowledge of the authors, they could largely help to improve the current unsatisfactory situation of models using a value, which is definitely not correct/representative, no matter what the scale of the model is.

4. The authors very often claim that secondary processes, e.g. bacterial methane production (or molecular diffusion) have occurred. However, this seems to be a (pure) speculation, because no measures of these processes are presented in the manuscript. These statements need more justification!

Specific comments:

Abstract:

Line 16: Here you promise " . . . this study provides representative d13C-CH4 signatures to be used in regional and global models . . ." is this really the case? See my general comment #3.

Line 19: "Progression in coal rank and secondary . . ." See my general comment #4

Introduction:

Line 38: "calculated" should perhaps better read "reported"

Line 48: Here the reference "Hein et al., 1997" should be added

Sec. 1.1. (there should be a section 1.2 if there is 1.1)

In general, in particular for a non-specialist, it would be more clear if the processes during coalification (many million years ago) and what the situation is today, when these coal deposits are sampled, would be discussed here separately.

First paragraph: I am a bit confused here – is it already called "coalification" when fatty acids are converted to CH4 ?

Line 87: should be "del13C" not "13C"

Line 108 – 112: Please clarify this sentence, do you mean ranges of 30/25 permil?

Line 117-118: See general comment #4: Has the theory (interaction with water) really been tested in the current study?

Section 2:

Line 143: Should read "km2" (not km)

Line 164: the word "origin" is missing

Line 183: Is it important to mention here the precision of d13C-CO2 measurements?

Line 185: Please add "CH4"

Line 197: Please give reference to the ArcGIS software (if these maps are not removed, as suggested above).

Line 204: Please be precise: It is not possible to measure mole fractions of emissions

Section 3: A lot of this detailed information may better go into a table.

Line 248: Do you really mean +33.3 permil?

Line 255: Please be precise: an emission cannot be "above background"

Lines 292 ff and line 310: please see general comment #4

---

## Author Response (AR1)

We thank the referees for the positive and very constructive comments. We have made the suggested changes accordingly and we have clarified the points that were raised in the comments below. The changes are highlighted in the second version of the manuscript that has been submitted.

**Anonymous Referee #1**

**Minor comments:**

*- One useful information that I think is missing is the importance of the methane emissions from the regions studied. Some values are provided for the Silesian mines, and some estimates of the coal production is given for the Australian ones, but how much does that represent on the world's scale, and how important are the British mines? Some literature references would suffice.*

Literature values for the relative contribution to global emissions at global scale of the coal basins surveyed have been provided.

See Lines 122-123, 144-145, 163-164.

*- Table 2 could be extended with information on the mine type (open, deep, active, inactive since, . . ., perhaps also coordinates) and coal type.*

Table 2 cannot be extended with such information, as most of the isotopic values reported in literature were obtained by sampling CBM (Coalbed Methane) wells across the basin and no specific mine is reported in these studies. Table 1 has been extended instead, to include information on type of mine and coal.

*- I don't find the map in Figure 1 very interesting: showing just the location of the mines without context information does not bring much to the paper. You could add some simplified geological map, or a map of the estimated distribution of coal mining methane emissions. Also, it is strange not to have the Australian and Silesian mines on a map!*

Figure 1 has been removed, and Figure 2 includes now location of the Hunter Coalfield.

*- Are you expecting any long or short-term variability of the d13C signature within one coal mine? If you were to re-do the sampling at a different season, or in a few years, would you expect any different result?*

After closure coal mines continue to emit methane into the atmosphere through breakages in the coal seam or underground conduits (i.e. wells, vent pipes), until they are entirely flooded due to the groundwater inflow. The rate of methane released after the closure tends to decline over the years. Furthermore, the intrusion of water might trigger biogenic production of methane, which will make emissions lighter ($^{13}$C-depleted). However, this assumption could not be tested, as surveys were carried out only once for most of coal mines, except for Maltby and Hatfield that were surveyed twice in the same year, for which the isotopic composition of methane emissions was really consistent.

*- Many of the readers of this paper won't be geologists. Although Section 1.1 is helpful and well written, it may help to provide some diagram showing the coal maturation process (when to expect which coal type), to be used as a reference when reading the following sections.*

A diagram showing the coal maturation process has been added (Figure 1).

**Textual comments:**

*- Line 26-27: what about Germany*

The shift from coal to natural gas involved most of Europe (Line 28)

*- Line 67: . . . methanogenic path: acetoclastic . . . (semicolumn missing)*

Corrected – see Line 68

*- Line 255: the measurements are not significantly above background: is that because the mine emits less, or because the emissions are spread over a larger area in such an open mine?*

In surface mines emissions are spread over a wider area as they are not focused on ventilation shafts.

*- L298: why would the depth affect the d13C (other than the depth/coal-rank relation?)*

The migration of methane towards the surface might lead to a fractionation effect as explained in the introduction. Line 304-305

*- L310: There is a 20-35 permil difference between the Hunter Coalfield and the Wales mines, and you are talking of a 5-10 permil shift here . . .*

The 5-10 permil difference refers to the difference between emissions from surface and deep mines within the same coalfield.

*- L321-323: Do you have any information on the proportion of each coal/mine type in the world?*

Added – see Line 334.

**I. Levin (Referee)**

*1. The results are currently presented in a very descriptive way, giving a lot of detail that makes part of the manuscript hard to read. I feel that a lot of this text, e.g. about locations, setting, would better fit into Table 1.*

Most of this text has been moved into Table 1, which now includes the type of coal and the coal seam for each of the mine surveyed.

*2. For each campaign, the authors report on the maximum CH4 signal and present detailed figures about the measured concentration distributions. However, this information is not evaluated in any way in the manuscript (e.g. by applying a dispersion model to estimate emission rates). The concentration signals largely depend on the actual meteorological situation at the time of sampling; therefore, a value of e.g. 10 ppm, which certainly confirms that there are indeed releases from the mining area, does not tell anything quantitative about the strength of the emission. I therefore feel that most of the figures are not required. Instead, in order to give the reader a feeling about the homogeneity of the Keeling-plot derived isotopic signature at each individual location and campaign, it would be more appropriate to present, perhaps in one multi-panel figure, individual Keeling plots from ALL campaigns, therewith showing the individual data points and regression lines in a standardized way (e.g. similar as Figure 5c, may be indicating also the maximum concentration in addition to 1/CH4).*

A multi-panel figure of Keeling plots of all campaigns has been added (Figure 3). All mole fraction maps were removed except for the map of the Hunter Coalfield, as in this case the location of sampled emissions from deep mines are essential for the interpretation of results.

*3. As also indicated by referee #1, I would strongly recommend to prepare an additional figure that summarizes all results, may be including the already published values from Table 2, in a diagram showing the relation between isotopic signature and coal type. This would be of tremendous help for people who want to use these results in their (modelling) studies. Currently, the last sentence in the Abstract leaves the reader/user somehow alone with the problem that rather no representative or at least improved input value for d13C of CH4 from coal is available. It looks to me that with the background knowledge of the authors, they could largely help to improve the current unsatisfactory situation of models using a value, which is definitely not correct/representative, no matter what the scale of the model is.*

A diagram showing the process of coalification and the relation between isotopic signature and coal type has been added. (Figure 1).

*4. The authors very often claim that secondary processes, e.g. bacterial methane production (or molecular diffusion) have occurred. However, this seems to be a (pure) speculation, because no measures of these processes are presented in the manuscript. These statements need more justification!*

The theory of secondary processes leading to more [13]C-depleted emissions is based on previous studies of many coal basins that are cited in the introductory section (Line 85-86; 100-103). The correlation of the isotopic composition with the biogenic production triggered by the incursion of meteoric water is an assumption based on measurements found in literature, but fits well with our results. However, such correlation will be presented in this paper only as a hypothetical source of variability in the isotopic composition of emissions from both surface and underground mines in the same coal basin (Line 299).

**Specific comments:**

**Abstract:**

*Line 16: Here you promise " . . . this study provides representative d13C-CH4 signatures to be used in regional and global models . . ." is this really the case? See my general comment #3.*

The sentence has been changed (Line 17).

*Line 19: "Progression in coal rank and secondary . . ." See my general comment #4*

Secondary processes are indeed suggested as a source of variability in the isotopic composition of coal-derived emissions, even though this assumption is based on the comparison of our results with those of other studies and not on direct measurements of secondary biogenic activities.

**Introduction:**

*Line 38: "calculated" should perhaps better read "reported"*

Changed – see Line 38

*Line 48: Here the reference "Hein et al., 1997" should be added*

Added – see Line 48

***Sec. 1.1.*** *(there should be a section 1.2 if there is 1.1).*

*In general, in particular for a non-specialist, it would be more clear if the processes during coalification (many million years ago) and what the situation is today, when these coal deposits are sampled, would be discussed here separately.*

Section 1.1 has been incorporated in the introduction.

The current geological situation differs between coal basins and was included in the description of the single basins in section 2.1. Section 1.1. aims to provide a general understanding of those processes that affect the isotopic composition of methane, that can have occurred directly during the process of coalification itself or in a second stage.

*First paragraph: I am a bit confused here – is it already called "coalification" when fatty acids are converted to CH4 ?*

Yes, it is. The decomposition of organic matter that leads to the peat formation is the first stage of the coalification process.

*Line 87: should be "del13C" not "13C"*

Changed – see Line 88

*Line 108 – 112: Please clarify this sentence, do you mean ranges of 30/25 permil?*

Clarified – see Line 109

*Line 117-118: See general comment #4: Has the theory (interaction with water) really been tested in the current study?*

No and this has been removed from Line 118.

**Section 2:**

*Line 143: Should read "km2" (not km)*

Corrected – see Line 144

*Line 164: the word "origin" is missing*

Added – see Line 167

*Line 183: Is it important to mention here the precision of d13C-CO2 measurements?*

Precision has been removed. Line 186

*Line 185: Please add "CH4"*

Added – see Line 188

*Line 197: Please give reference to the ArcGIS software (if these maps are not removed, as suggested above).*

The source of the GIS map has been added in the caption of Figure 2

*Line 204: Please be precise: It is not possible to measure mole fractions of emissions*

The sentence has been rephrased. Line 210-215

**Section 3***: A lot of this detailed information may better go into a table.*

Much of this information is now included in Table 1

*Line 248: Do you really mean +33.3 permil?*

Thank you – corrected on Line 252

*Line 255: Please be precise: an emission cannot be "above background"*

Corrected – see Line 259

*Lines 292 ff and line 310: please see general comment #4*

As the likelihood of biogenic secondary methane cannot be proven in the present study the statements on Lines 298, 318 are now only suggestions.

[revised manuscript text omitted]

---

## Author Response (AR2)

Dear Dr. Ingeborg Levin, thank you very much for your suggestions. We have made the technical changes suggested. You can find the changes in the third version of the manuscript that has been submitted.

**Technical corrections**

**- Figure 1 would largely benefit from some more quantitative data. The current information could easily be given in one sentence of the text without an extra figure.**

Quantitative data (i.e. temperature, burial pressure, C content and methane isotopic signatures) have been added. Please note that isotopic ranges for coal-derived methane emissions are based on the isotopic values on Table 2. Therefore, Figure 1 has been moved to the discussion section and it is now Figure 3 (see line 525).

**In Figure 3 please add y-axes titles (del13C) in all "lines" and an x-axis title (1/CH4) in the second column (4th line.)**

Titles have been added.

[revised manuscript text omitted]

Figure 3 shows the $\delta^{13}CCH_4$ signatures ranges for each coal type, based on the results of this study and isotopic values found in literature.

**4    Conclusions**

By measuring the isotopic signatures of methane plumes from a representative spread of coal types and depths, we show that the $\delta^{13}$C isotopic value to be included in regional and global atmospheric models for the estimate of methane emissions from the coal sector must be chosen according to coal rank and type of mining (opencast or underground). For low resolution methane modelling studies, our measurements suggest an averaged value of -65 ‰ for bituminous coal exploited in open cast mines and of -55 ‰ or less negative, in deep mines, whereas values of -40 ‰ and -30 ‰ can be assigned to anthracite opencast and deep mines respectively. However, the ranges of isotopic signatures of methane emitted from each of these categories are wide (see Fig. 3), and the isotopic ratios of methane from Chinese coal mines is little studied. Further measurements would be required in other coal mining areas, especially China, to determine appropriate values for global modelling of methane isotopes.

[revised manuscript text omitted]

Yao, Y., D. Liu and Y. Qiu, 2013, Variable gas content, saturation, and accumulation characteristics of Weibei
coalbed methane pilot-production field in the southeastern Ordos Basin, China: AAPG bulletin, v. 97, p.
1371-1393.

Zazzeri, G., Lowry, D., Fisher, R., France, J., Lanoisellé, M., and Nisbet, E., 2015, Plume mapping and isotopic
characterisation of anthropogenic methane sources: Atmospheric Environment, v. 110, p. 151-162.

[revised manuscript text omitted]

**Figure 3 Coal maturation process. The $\delta^{13}C$CH$_4$ signatures ranges are based on the literature values in Table 2 and on the results of this study.**